# British Gynaecological Cancer Society Recommendations for Evidence Based, Population Data Derived Quality Performance Indicators for Ovarian Cancer

**DOI:** 10.3390/cancers15020337

**Published:** 2023-01-04

**Authors:** Sudha Sundar, Andy Nordin, Jo Morrison, Nick Wood, Sadaf Ghaem-Maghami, Jo Nieto, Andrew Phillips, John Butler, Kevin Burton, Rob Gornall, Stephen Dobbs, Rosalind Glasspool, Richard Peevor, Jonathan Ledermann, Iain McNeish, Nithya Ratnavelu, Tim Duncan, Jonathan Frost, Kenneth Lim, Agnieszka Michael, Elly Brockbank, Ketankumar Gajjar, Alexandra Taylor, Rebecca Bowen, Adrian Andreou, Raji Ganesan, Shibani Nicum, Richard Edmondson, Richard Clayton, Janos Balega, Phil Rolland, Hilary Maxwell, Christina Fotopoulou

**Affiliations:** 1Pan Birmingham Gynaecological Cancer Centre, City Hospital and Institute of Cancer and Genomic Sciences, University of Birmingham, Birmingham B152TT, UK; 2East Kent Hospitals University Foundation NHS Trust, Ethelburt Road, Canterbury CT1 3NG, UK; 3National Cancer Registration & Analysis Service (NCRAS), NHS Digital, Wellington House, 133-155 Waterloo Rd., London SE1 8UG, UK; 4GRACE Centre, Department of Gynaecological Oncology, Musgrove Park Hospital, Somerset NHS Foundation Trust, Taunton TA1 5DA, UK; 5Lancashire Teaching Hospitals NHS Foundation Trust, Preston PR7 1PP, UK; 6Department of Surgery and Cancer, Imperial College, London W12 0NN, UK; 7Norfolk and Norwich University Hospital, Colney Lane, Norwich NR4 7UY, UK; 8Derby Gynaecological Cancer Centre, University Hospitals of Derby and Burton, Royal Derby Hospital, Uttoxeter Rd., Derby DE22 3NE, UK; 9Royal Marsden NHS Foundation Trust, London SW3 6JJ, UK; 10Glasgow Royal Infirmary, PRMH Building, Glasgow G4 0SF, UK; 11Department Gynaecological Oncology, Cheltenham General Hospital, Gloucestershire Hospitals NHS Foundation Trust, Gloucester GL53 7AN, UK; 12Department of Gynaecological Oncology, Belfast City Hospital, Belfast BT9 7AB, UK; 13Beatson West of Scotland Cancer Centre, NHS Greater Glasgow and Clyde and University of Glasgow, Glasgow G12 0YN, UK; 14Consultant Gynaecological Oncologist, Clinical Lead for Colposcopy & Gynaecological Cancer MDT, Betsi Cadwaladr University Health Board, Bangor LL57 2PW, UK; 15UCL Cancer Institute, University College London and UCL Hospitals, London WC1E 6DD, UK; 16Northern Gynaecological Oncology Centre, Queen Elizabeth Hospital, Gateshead NE9 6SX, UK; 17Royal United Hospitals Bath NHS Foundation Trust, Bath BA1 3NG, UK; 18Cardiff and Vale UHB, Cardiff CF14 4XW, UK; 19University of Surrey and Royal Surrey County Hospital, Guildford GU2 7XH, UK; 20Royal London Hospital, Barts Health NHS Trust, London EC1 1BB, UK; 21Nottingham University Hospitals NHS Trust, City Hospital, Nottingham NG5 1PB, UK; 22Department of Radiology, Royal United Hospitals Bath NHS Foundation Trust, Bath BA1 3NG, UK; 23Department of Cellular Pathology, Birmingham Women’s Hospital, Birmingham SY16 4LE, UK; 24Division of Molecular and Clinical Cancer Sciences, School of Medical Sciences, Faculty of Biology, Medicine and Health, The University of Manchester and St Mary’s Hospital, Manchester M13 9PL, UK; 25Manchester University NHS Foundation Trust, Manchester M13 9WL, UK; 26Pan Birmingham Gynaecological Cancer Centre, City Hospital, West Midlands, Birmingham B15 2SQ, UK; 27Dorset County Hospital NHS Foundation Trust, Dorset DT1 2JY, UK; 28Department of Surgery and Cancer, Gynaecologic Oncology, Imperial College London, London W12 0NN, UK

**Keywords:** quality performance indicators, British Gynaecological cancer society, ovarian cancer, population data

## Abstract

**Simple Summary:**

Ovarian cancer survival in the UK is poorer than other similar countries. Results from the National Ovarian Cancer Audit Feasibility Pilot (OCAFP) showed that approximately 1 in 4 women with advanced stage ovarian cancer (greater than Stage 2) do not receive any anti-cancer treatment and that only 51% will receive both surgery and chemotherapy in England. The audit also showed that the proportions of women receiving treatment varies a lot across different areas in England. In response, a multidisciplinary team from the British Gynaecological cancer society has established Quality performance indicators that can be evaluated regularly using routinely collected data and will help improve cancer outcomes.

**Abstract:**

Ovarian cancer survival in the UK lags behind comparable countries. Results from the ongoing National Ovarian Cancer Audit feasibility pilot (OCAFP) show that approximately 1 in 4 women with advanced ovarian cancer (Stage 2, 3, 4 and unstaged cancer) do not receive any anticancer treatment and only 51% in England receive international standard of care treatment, i.e., the combination of surgery and chemotherapy. The audit has also demonstrated wide variation in the percentage of women receiving anticancer treatment for advanced ovarian cancer, be it surgery or chemotherapy across the 19 geographical regions for organisation of cancer delivery (Cancer Alliances). Receipt of treatment also correlates with survival: 5 year Cancer survival varies from 28.6% to 49.6% across England. Here, we take a systems wide approach encompassing both diagnostic pathways and cancer treatment, derived from the whole cohort of women with ovarian cancer to set out recommendations and quality performance indicators (QPI). A multidisciplinary panel established by the British Gynaecological Cancer Society carefully identified QPI against criteria: metrics selected were those easily evaluable nationally using routinely available data and where there was a clear evidence base to support interventions. These QPI will be valuable to other taxpayer funded systems with national data collection mechanisms and are to our knowledge the only population level data derived standards in ovarian cancer. We also identify interventions for Best practice and Research recommendations.

## 1. Background

Implementation of centralised care in oncology has been a decisive turn in the management of cancer patients worldwide [1]. The United Kingdom (UK) was one of the first countries to implement such an approach, starting in the 1990s in an effort to improve patient experience, access to specialised care and ultimately low survival in the UK [2,3,4]. The Calman–Hine report was the first comprehensive cancer report to be produced in the UK, which set out the principles for cancer care and re-organisation of clinical service delivery, advocating a change from a generalist model towards a fully specialist service [5]. Gynaecological cancer care operates on a hub and spoke model in the UK, with secondary care facilities, called cancer ‘units’, providing diagnostic services and tertiary care facilities, called ‘centres’, providing specialist cancer surgery, systemic treatment and radiotherapy. Health care in the UK is publicly funded; thus in theory resource allocation should be equitable for all patients and across all geographical regions.

Despite centralisation, multiple data over the past years raise significant concerns about ovarian cancer care in the UK. Particular concerns are around those patients with widespread advanced, or relapsed, disease who require specialised abdominal surgery. Limited infrastructure resources, including operating theatre time, intensive care capacity, gaps in surgical skills training, workforce pressures, and diagnostic delays, are key contributors to suboptimal care despite the implementation of a centralised model [6,7,8,9,10]. The effect of the limitations are reflected in ovarian cancer survival statistics, with the UK lagging behind comparable countries [11]. With a five-year overall survival of 36.7% (95% CI 36.1 to 37.4), UK data were 9.1% below Norway, the best performing of the nations reported. Age-standardized one-year survival from ovarian cancer was 69.4% in the UK, compared with 77.4% to 76.6% in Australia, Denmark and Norway [12]. Survival in the UK is particularly poor for older patients and those with advanced stage ovarian cancer; in patients with advanced stage disease aged 65–74, three-year survival in the UK was 33% compared to 52% in Norway [12]. For women diagnosed between 2010–2014, 5 year survival in England is 37% compared to 32% in Northern Ireland, Scotland and Wales.

These data led to the development of a nationwide audit, funded jointly by the British Gynaecological Cancer Society (BGCS) and two major UK-based ovarian cancer charities (Target Ovarian Cancer and Ovarian Cancer Action), to investigate the reasons underlying poor survival, and to develop strategies to overcome them [13]. The National Ovarian Cancer Audit feasibility pilot (OCAFP) utilised routinely collected cancer registry data within the National Cancer Registration and Analysis Service (NCRAS) to analyse outcomes for all patients with ovarian cancer in England [10,13]. OCAFP demonstrated major deficiencies in the management of the ovarian cancer patients in England, with approximately one in four women with FIGO Stage II-IV ovarian cancer [14], not receiving any anticancer treatment, either surgical, or systemic chemotherapy. Moreover, only 51% of patients received international standard of care treatment, i.e., the combination of surgery and chemotherapy [13]. The significant variation in the rate of patients with advanced ovarian cancer who received anticancer treatment across Cancer Alliances (the 19 geographical regions in England for organisation of cancer delivery) is a further crucial finding of the OCAFP. While stressing differences in the time coverage, covariate adjustment and cohort definitions of the two reports, analyses indicate that Cancer Alliances that were less likely to undertake treatment generally had lower than average five year survival figures, and that this relationship may be particularly pronounced for surgery. These differences in treatment correlate with five-year net survival rates across England ranging from 28.6% to 49.6% [13] (Appendix A).

Whilst individual cancer centres may report high treatment rates and good survival in England, these national population-based data are sobering. This is however, not unique to England. Population-based data from the United States (US) revealed that 21% of ovarian cancer patients in the American National Cancer Database and 34.2% in the Surveillance, Epidemiology, and End Results (SEER) database did not undergo surgery [15,16]. Similarly, another population-based study from the US demonstrated that only 37% of ovarian cancer patients received guideline-compliant surgery and chemotherapy [17]. Interestingly, the regional variation in survival seen in the OCAFP have also been demonstrated in other countries with similar health care systems [18].

In response to the findings of the OCAFP, the BGCS convened a multidisciplinary panel comprising elected BGCS regional representatives, key opinion leaders in gynaecological medical and surgical oncology, as well as representatives from cancer charities, NCRAS, NHS England Specialist Commissioning and the National Cancer Research Institute (NCRI). An open consultation exercise with the gynaecological cancer community in the UK was undertaken to secure community engagement, applicability, and nationwide endorsement. This manuscript summarises ongoing efforts to address the audit results through development of evidence-based, consensually agreed, quality performance indicators (QPIs) and provides an example of how a clinical community can mobilise and effect change in a taxpayer funded health system. We also discuss implementation efforts and broader initiatives to address system gaps.

In summary, there is strong evidence in the UK to show significant variability, both in ovarian cancer survival and in treatment rates. This is despite a centralised system of care where only designated cancer centres can provide surgery for ovarian cancer. Here we consider data driven QPIs as a potential driver for improvement.

## 2. Learning from Other Health Systems: The Case for Evidence-Based, Population Data Derived, Routinely Evaluated QPIs

QPIs, a set of robustly derived, regularly audited indicators across domains, are a useful way to assess performance, monitor progress towards goals, and compare results across organisations [19]. They enable identification of best practice and can support shared learning. A metrics-based approach has been established to accredit and audit cancer centres in other tumour types, such as pancreatic, rectal and urological cancers, in healthcare systems equivalent to the UK, including Canada, Australia and Portugal [20,21,22,23]. QPIs include both clinical outcome measures, such as rate of postoperative complications, surgical mortality, surgical resection margins, colostomy rates, length of stay, readmission, recurrence rates and overall survival, as well as process measures, such as time from diagnosis to complete staging and to start of treatment.

The European Society of Gynaecological Oncology (ESGO) has set out ESGO QPIs for the accreditation of ovarian cancer centres in a two-tier system: simple accreditation and centre of excellence [24]. As a result, substantial effort has occurred in Europe, and beyond, to improve performance at a centre-level to secure ESGO accreditation. Furthermore, governments have used these metrics for approval of funding and commissioning of individual centres and healthcare systems. Increasingly, patients drive this process by seeking second opinions from accredited centres as they are easily identifiable online.

However, both ESGO QPIs and international ovarian cancer treatment guidelines focus on improving treatment standards mainly at the level of individual hospitals, without taking into consideration the overall infrastructure, commissioning, and service delivery models of the entire healthcare system of each country [24,25,26]. Centre-specific guidelines and metrics do not address key health system aspects that affect implementation of optimal care, such as access for all patients to specialised care, financial sustainability at individual, as well as health system level, and referral patterns where patients are often triaged prior to attending a cancer centre or diagnostic pathways [27]. Whilst such quality performance metrics are useful for individual cancer centres, they do not compare ‘like with like’, are not based on ‘whole patient cohorts’, do not take account of all patients affected with cancer in a nationwide healthcare system, and provide a selective view of cancer outcomes, as referral bias is implicit. Additionally, the metrics are often based on ‘expert opinion’.

Population-based data differs from centre-submitted data in several key aspects; population-based data includes all patients in a defined geography within a specified time, includes patients who are elderly, multimorbid, those with advanced or have untreatable disease, regardless of the individual’s ability to afford care, and without bias from triage, based on referral criteria to a specific centre [28]. These datasets should ideally be collected routinely from every patient with the condition within a defined population. This contrasts with centre-submitted and clinical trial data that has the potential to select patients with better performance status/better socioeconomic status to undergo more extensive treatment. Population-based data provide a more accurate picture of cancer outcomes for the general population, particularly with robust cancer registration at a whole population level, as in England. Deriving QPI from population-based data, that is collected routinely, facilitates implementation of quality metrics into practice and assessment of comparative performance whilst limiting selection bias.

For instance, Scotland has developed a set of tumour site-specific and generic national QPIs and performance against these indicators is collected at a population level. Scottish Ovarian Cancer QPIs have led to diagnostic pathway improvements, improvements in surgery rates and outcomes. Robust data collection and clear data definitions have also enabled comparison between Scottish centres and benchmarking against international population-based data. Collection of the second cohort of survival data is ongoing and will demonstrate if service improvements, based on Scottish QPIs, have led to improved survival.

In summary, we identify the need to establish QPI for ovarian cancer that are relevant to the UK NHS and that are population-based and evaluable from routinely collected cancer registry data. This contrasts with ESGO QPI which are based on hospital submitted data.

## 3. Principles Underlying Identification of Metrics as QPIs from Population-Based Data

The principles underlying the selection of QPIs included: A clear definition; an evidence-based approach as derived from the OCAFP [13]; and that required datasets would be readily available through routine data collection systems via NCRAS. Performance against metrics will be displayed in public domain at the level of Cancer Alliances and cancer centre-unit combinations. Hospital trusts/individual cancer service providers would have access to hospital level, granular data behind the NHS firewall on the ‘CancerStats2′ platform to protect patient confidentiality [29]. This pragmatic approach was viewed as the most sustainable way for the smooth implementation of QPIs into routine clinical practice and monitoring of performance. The CancerStats2 platform is accessible to NHS employees, and it is anticipated that demonstrating data at hospital trust level and dissemination of these reports to Cancer Alliances and hospital trust senior management will prompt a greater level of scrutiny. This allows Cancer Alliances to review their performance against metrics regularly and enables development of quality improvement initiatives. Figure 1 is a schematic of development, Table 1 provides a summary of all QPI with definitions, numerators and denominators.

Standard of care for first line treatment in ovarian cancer care involves a combination of surgical procedures, to achieve macroscopic cytoreduction (removal of all visible disease), platinum-based combination chemotherapy, with or without the addition of bevacizumab, and genomic testing to guide maintenance therapy with PARP inhibitors to improve duration of remission [26,30,31]. Performance metrics would assess compliance against these key factors, in addition to process and structural indicators.

The term ‘ovarian cancer’ is used in a collective manner to include all ovarian, fallopian tube and gynaecological primary peritoneal cancers. Definitions of key terms, such as diagnosis, anticancer treatment and methodology, are as set out in the OCAFP [13]. All analyses of performance against QPIs in advanced stage ovarian cancer (FIGO stage II–IV) [14] to be presented adjusted for age/stage/histology type/comorbidity/deprivation. Thus, Cancer Alliances with older populations will not be disadvantaged. Further indices for adjustment of performance will include WHO Performance Status [32] and BRCA germline status, if data capture is sufficient.

### 3.1. QPI 1: Patients to Be Discussed at Diagnosis at a Specialist Multidisciplinary Team (MDT) Prior to a Decision for Treatment; Target 95%

#### 3.1.1. Rationale

The OCAFP has shown that nearly one in four women with advanced ovarian cancer do not receive any anticancer treatment with significant variations across the Cancer Alliances after adjustment for age and deprivation factors [13]. The reasons for this are multifactorial and need further evaluation, but may be attributed to a patient’s poor performance status at presentation, often related to a higher tumour volume and delayed diagnosis.

Analysis of early mortality at the OCAFP demonstrates that patients diagnosed at cancer units have about 10% lower survival than patients diagnosed at cancer centres [13]. The reason for this is unclear and needs further research. One hypothesis is that poorer outcomes result from a longer diagnostic process, particularly in older, more frail patients. Evidence shows that patients with cancer managed by subspecialists in gynaecological oncology and by MDTs have better outcomes [33,34]. It is hoped that establishing this QPI results in a heightened focus on diagnostic pathways across community, primary, secondary and tertiary care service providers. Discussing all patients means that decisions are made by the specialist MDT, rather than excluding some patients from this process, who may have decisions made by individual clinicians. It also allows collection of more complete data, to better understand why no treatment was offered. Whilst not included as a metric, an MDT discussion across all critical decision-making points in a treatment pathway is strongly encouraged, so that patients benefit from a multidisciplinary input across all stages of their cancer: from diagnosis to palliation.

#### 3.1.2. Best Practice Solutions

NHS England has set out guidance for timed diagnostic pathways, aiming to achieve diagnosis within 28 days of a GP referral [35]. This guidance recommends establishment of ‘one stop’ clinics for patients referred with symptoms or signs of ovarian cancer. These clinics will have rapid access to imaging, imaging-guided biopsy, comprehensive assessment of fitness and ‘prehabilitation’ advice for frailer patients. The BGCS supports establishment of these clinics as best practice. Approximately one in four patients with ovarian cancer are diagnosed as emergency admissions, which is associated with very poor outcomes [36]. Rapid diagnostic clinics (RDCs) and community diagnostic hubs are NHS initiatives designed to promote earlier diagnosis [37]. Research into optimal diagnostic pathways in primary care, as well as in secondary care is ongoing [38,39].

### 3.2. QPI 2: Patients Diagnosed with FIGO Stage II-IV or Unstaged Ovarian Cancer to Receive Anticancer Treatment (of Any Type); Target 80%

#### 3.2.1. Rationale

The OCAFP shows that on average 73% of women with advanced ovarian cancer receive any anticancer treatment (model adjusted for age/performance status/deprivation) [13]. Eight of the 19 Cancer Alliances currently fall below this average, with five falling more than two standard deviations (SD) below the average. The QPI assumes that all patients with Stage I disease, with very rare exceptions, will receive treatment.

Women who do not receive anticancer treatment have a much worse prognosis, and whilst it will be appropriate for some patients not to receive treatment, the large variations across England suggest that it is not just patient-factors that result in patients not receiving treatment. Patients need to be adequately informed about all available treatment options, the associated risks and benefits, and the consequences of omission of standard of care, before they reach a decision to decline any anticancer treatment. Not receiving systemic anticancer treatment will be appropriate for some patients, depending on their wishes and performance status; a population-based study from the Netherlands has shown that 14 to 20% of patients did not receive anticancer treatment [40].

#### 3.2.2. Best Practice Solutions

The BGCS encourages cancer services providers to incorporate systematic assessment of frailty, so that fitness for surgical and systemic therapy can be assessed in a consistent manner. One factor that may be amenable to improvement within secondary care is ensuring that patients in non-specialist wards are assessed promptly and reviewed by acute oncology teams or gynaecological oncology specialist nurses to start this process. The BGCS anticipates that this QPI will promote closing working relationships between trust-wide acute oncology services and tertiary-level Gynaecological oncology services (surgeons, oncologists, clinical nurse specialist teams). Efforts to disseminate awareness of ovarian cancer in women presenting as a medical emergency to other specialties, particularly with non-specific symptoms, will be vital. Implementing reflex cytological testing of ascites in previously undiagnosed patients, and standardised radiological assessment to assess disease extent, should be standard practice. More research is needed to understand why patients do not receive treatment, including to explore patients’ preferences.

### 3.3. QPI 3: Patients with FIGO Stage II-IV/Unstaged Ovarian Cancer to Receive Cytoreductive Surgery. Target: Minimum Target 55%; Optimal Target 70%

#### 3.3.1. Rationale

The OCAFP shows that on average only 51% of women with FIGO Stage II-IV and unstaged ovarian cancer will receive cytoreductive surgery in England in the nine months following diagnosis [13]. Four out of the 19 Cancer Alliances lie more than two SD below the national mean intercept, with less than half of their patients undergoing surgery. There is ample evidence that patients who have surgery as part of their treatment package will have a more favourable outcome than those patients treated with chemotherapy alone [15,41]. *Post hoc* analysis from the ICON8 trial clearly demonstrates that even those patients with RECIST stable disease after neoadjuvant chemotherapy benefit from surgical debulking [42].

We appreciate that a minimum standard for surgery at 55% appears too low; we strongly emphasise that this represents 55% of the whole cohort of women with advanced cancer. This cannot be compared to trial data or institutional figures which are subject to case selection bias. Our data is in keeping with population data both from the UK [43] and Nordic countries. In setting this target, the BGCS panel considered the fact that a proportion of individual Cancer Centres will have performance well above, as well as some below, the national mean intercept for Cancer Alliances. Thus, standards were set to promote higher standards, whilst acknowledging the efforts and resources needed to achieve these targets, in some Cancer Alliances, will be substantial.

#### 3.3.2. Best Practice Solutions

All patients with a presumed or confirmed diagnosis of ovarian cancer should be seen and assessed within an MDT structure to be allocated to the appropriate treatment pathways, be that upfront surgery, neoadjuvant chemotherapy (NACT), primary chemotherapy or palliative care alone. Patients who receive NACT should be re-evaluated after three to four cycles within the MDT to confirm their eligibility for surgery and subsequent treatment plan. The BGCS has, in collaboration with the national colorectal, upper gastrointestinal surgical and general surgical societies, established a framework for joint operating and governance, which is expected to support gynaecological oncologists in the delivery of maximal effort ovarian cancer surgery [44].

### 3.4. QPI 4a: Patients with Ovarian Cancer Should Have Recording of FIGO Stage and WHO Performance Status at Initial Presentation; Target 95%. QPI 4b: Patients with Ovarian Cancer Who Undergo Cytoreductive Surgery Should Have the Quantity of Postoperative Residual Disease Recorded; Target 95%

#### 3.4.1. Rationale

Understanding decision-making and treatment variation across providers is enhanced by accurate patient and tumour-related data. Recording of FIGO stage at presentation varies between Cancer Alliances in England, which may reflect efforts made by MDTs to stage accurately and carefully consider patients for anticancer therapy. Currently 65% of patients with FIGO Stage II–IV/unstaged ovarian cancer had missing data on performance status [13,45].

Data completeness of residual disease within Cancer Outcome Service Data (COSD) is currently poor. This greatly limits any effort to understand surgical practice or address surgical skills training in England. Systematic description of tumour dissemination patterns, highest tumour load, and site and size of postoperative residual disease, reasons why disease was unresectable at the time of surgery, performance status and disease distribution to assign surgical stage, are of paramount importance to characterise adequately the tumour profile, estimate prognosis and tailor adjuvant treatment.

#### 3.4.2. Best Practice Solutions

Operation notes should be compiled in a way that any third professional party, not present at the operation, could follow the thought process of the surgeon and understand the rationale of the procedures performed and the outcome. Templates to record this may be useful and help standardisation, an example from the Royal Marsden Hospitals is displayed as Appendix A. The BGCS encourages compiling of electronic operation notes in a standardised format. National data collection systems, e.g., Somerset Cancer Register and Infoflex, which capture the cancer outcomes and services datasets, have been modified to include residual disease status. MDT leads are encouraged to ensure that this is recorded for every ovarian cancer patient undergoing surgery. Comparative data on completion rates by cancer centre are regularly circulated to cancer service providers and individual clinicians, to promote compliance with this measure. The BGCS has instituted training initiatives for clinicians providing diagnostic services, enhancing surgical training and funding fellowships for travel to enhance skills across all specialties involved in gynaecological cancer care (BGCS-Basingstoke fellowship; BGCS cancer unit study day; BGCS-awards-grants-and-prizes).

### 3.5. QPI 5a: Patients with Non-Mucinous Epithelial Ovarian Cancer to Be Tested for Germline BRCA1/2 Testing; Target 90%. QPI 5b: Patients with Advanced High Grade Serous and Clear Cell Carcinoma on Histology to Be Tested for Tumour BRCA1/2 Testing; Target 90%

#### 3.5.1. Rationale

Germline *BRCA1/2* testing identifies patients who carry an inheritable pathological variant (mutation) in *BRCA1/2* (*BRCA*mut) [10]. Data show that almost 40% of patients who carry a *BRCA*mut at their initial ovarian cancer diagnosis will have a negative family history [46]. Family history and age should not be used as guide to whom should be offered genetic testing [10]. There are significant therapeutic implications for patients with an underlying *BRCA*mut and a significant progression-free survival benefit for patients with *BRCA*mut from PARP-inhibitors [31]. Additionally, testing offers the opportunity for cascade testing of family members, thus giving the opportunity for preventative treatment for both breast and ovarian cancer. In setting this target, the BGCS acknowledges that testing may be declined by some patients and that the proportion tested may vary across populations. Research is needed to understand whether cultural and societal barriers exist to promoting uptake of *BRCA* testing and how these may be overcome. Culturally specific interventions may be necessary to facilitate uptake of testing.

Germline BRCA testing is provided free to the patient by the National Health Service.

Sustainability of quality health care in the context of rising costs is critical; in the UK new intervention or diagnostics are usually assessed for cost effectiveness by an arm’s length body called National Institute of Health and Care Excellence (NICE) and only implemented into routine health care if found to be cost effective [47].

#### 3.5.2. Best Practice Solutions

The BGCS has set out guidance encouraging parallel germline and tumour testing for *BRCA*mut testing patients with high grade serous and clear cell ovarian carcinoma [10]. A revision of guidance incorporating HRD testing will be released shortly. Work within NCRAS is underway to identify patients in whom tumour *BRCA* testing was performed.

### 3.6. QPI 6: Patients to Be Enrolled into an NCRI Portfolio Study at Diagnosis or at Any Point during Treatment; Minimum Target 5%

#### 3.6.1. Rationale

Evidence strongly suggests that patients want to participate in research trials, those who do have better outcomes, and centres with greater research recruitment deliver better outcomes for patients [9]. Data on research participation are collected routinely through the collaborative research networks and can be assessed and reported. The BGCS encourages that MDT leads work with regional research champions, and the regional research delivery team, to identify potential trials and sites for patients.

Trusts and Research and Development departments that act collaboratively by signposting patients to recruiting trials at other sites, should be recognised and given credit for promoting access to clinical research. The UK NCRI Gynaecological Group has a rich portfolio of trials across a wide spectrum from basic, to translation and clinical research. Centres are encouraged to participate in these trials. It is a surrogate marker for expertise, commitment to high quality care and infrastructure.

#### 3.6.2. Best Practice Solutions

Working with the Royal College of Surgeons (RCS), the BGCS has appointed BGCS-RCS Surgical Specialty leads for research who are tasked with promoting research participation [48].

### 3.7. QPI Considered and Not Approved at This Point

The BGCS carefully considered setting a QPI around the degree of cytoreduction achieved, the proportion of women receiving primary surgery and the extent of surgery. These were rejected for consideration at this time for the following reasons:National data capture on residual disease is limited (currently around 80%), thus a metric for the completeness of cytoreduction is not reportable;Current evidence does not support any defined metrics on the proportion of women with advance disease (FIGO Stage IIIC and above) who should receive either primary surgery or delayed primary surgery after neoadjuvant chemotherapy. Four completed randomised controlled trials have demonstrated equivalence in overall survival [30]; results of ongoing randomised studies are awaited [49,50,51,52,53].

Work is underway within the OCAFP to assess the extent of variability in surgical practice, but this methodology is not yet validated. Thus, at this time, a metric on the extent of surgical radicality cannot be introduced.

## 4. Implementing Metrics in Practice

Since the publication of reports from OCAFP, NHS England has agreed to fund a long-term audit on ovarian cancer to start in October 2022. The QPIs outlined above will be used as reference metrics. Importantly for patients, as well as for hospital trusts, data in public domain will include comparative performance against these metrics by Cancer Alliance and cancer centre-unit combinations. We anticipate the first public report of performance data on these metrics will be available by the end of 2023/24. OCAFP reports have been regularly sent to Cancer Alliances inviting them to review comparative performance and this will continue when the long-term, nationally funded audit is established.

In the UK, higher specialist training in gynaecological oncological surgery is delivered through a rigorously supervised programme at centres accredited by the Royal College of Obstetricians and Gynaecologists, in partnership with the BGCS. From 2022, compliance with BGCS consensus QPI metrics is mandatory for any centres applying for accreditation or re-accreditation as a training centre for Gynaecological Oncology subspecialty training in the United Kingdom. This not only ensure that future generations of subspecialist gynaecological oncological surgeons will be trained in high quality centres, but also embeds a drive to deliver these QPIs in clinical practice. Going forwards, performance of cancer centres and alliances will be available in public domain. We anticipate that this transparency will drive improvements.

The actual governance, medico-legal and financial consequences of the lack of compliance with metrics has not yet been agreed with the commissioners and NHS England. Nevertheless, the BGCS will work with cancer charities, patient advocates and the clinical community to support sharing of best practice to achieve performance compliant with these QPIs. The BGCS is also working with NHS specialist commissioning to generate a dashboard of performance against these metrics.

## 5. Discussion

The OCAFP has shown significant variation in treatment rates across England. This variation persists even after adjustment of age, stage, histology type, comorbidity, and deprivation index, suggesting that this variation is not explainable by patient factors. Variation is therefore more likely to be due to inconsistencies in the health system, which should be amenable to improvement. The BGCS has sought to take a systems-wide approach, encompassing both diagnostic and therapeutic pathways, to set out robust nationwide recommendations and metrics for practice. It will be important for the healthcare system to identify best practice and work collaboratively, so that best practice can be shared and implemented. Key areas for service improvement and research are also outlined (see Appendix A).

An enhanced commitment to careful, comprehensive, high-quality prospective data collection will be pivotal to understanding differences in survival and instituting improvements in care. More research is needed to identify the contributors to the variation in MDT decision-making and treatments across England. This includes understanding more fully the differences in local organizational factors, such as staffing, skill mix, access to operating theatre time, intensive care support, postoperative nursing care and the accessibility of systemic treatments at diagnosis and recurrence; all of these are likely to play key roles. The BGCS is currently analysing data from a granular workforce survey of staff across all specialties involved in gynaecological cancer care, this will help inform resourcing [54].

Cancer Alliances will vary in measures needed to achieve compliance with these QPIs; for example, the West Midlands Cancer Alliance includes five separate cancer centers. A statutory gynaecological cancer operational delivery network across cancer centers has been proposed, as a mechanism to ensure that high quality services can be delivered and within-region variation in treatment and survival differences can be improved. The solution within another Cancer Alliance, with a smaller population and a single cancer center, may be very different. There are currently no nationally mandated cancer operational delivery networks, aside from radiotherapy.

Whilst treatment guidelines across high income countries are comparable, in practice substantial variation is reported even amongst those with similar healthcare systems (e.g., Denmark, Canada and Norway). The regional variation identified in the OCAFP has also been demonstrated in other countries with similar health systems [6,18]. Potentially, centralisation of care and implementation of QPI has the power to alleviate regional differences, if implemented correctly and accompanied by the necessary infrastructure and training. A study of more than 15,000 patients from the Netherlands Cancer Registry has shown that since centralization of surgical care in 2012, the variation between hospitals in the probability of undergoing cytoreductive surgery for patients with advanced OC resolved [55]. The Scandinavian countries are an exemplar demonstrating improved surgical cytoreduction and survival rates in advanced ovarian cancer patients, as a result of the introduction of centralisation of surgery in the recent years [56,57]. However, recent Scandinavian data demonstrate no change in overall survival with further centralisation and improvements in surgical radicality, suggesting that there is a limit to what can be achieved from centralization alone [58]. These data fully correspond also with the UK experience, with UK’s survival data lagging behind comparable countries, despite the UK being one of the first countries worldwide that centralized cancer care [11].

Denmark has instituted a high quality Danish Gynaecological Cancer Database, collecting data on all women with ovarian cancer treated at Danish hospitals since 2005 and has implemented QPIs across gynaecological cancers [59]. Data support the evidence in favour of centralizing treatment for complex and heterogeneous diseases, and the introduction of clearly defined QPI metrics to assess comparative performance, to improve survival [55,60]. National lung, bowel, prostate cancer audits have driven sustained improvements in cancer outcomes in the UK and we are optimistic that the same will be true in ovarian cancer [61,62,63].

Stellar international efforts to improve ovarian cancer outcomes include the American Society of Clinical Oncology recent guidance stratified by resource on the diagnosis and treatment of Ovarian cancer and the ESGO-ESMO consensus documents on ovarian cancer. The European Society of Gynaecological Oncology has also done outstanding work in articulating hospital level QPI wherein hospitals provide data to ESGO for accreditation on a regular basis. This effort has been extremely successful in promoting quality standards across Europe. BGCS [64,65] QPI are different from ESGO QPI in following critical aspects—BGCS QPI do not rely on individual hospitals providing data but are evaluable and will be reported from routinely collected national registry data and encompass both cancer units and cancer centres. BGCS QPI are derived from population level data (whole cohort of patients, not limited to those treated at specific hospitals). Our efforts are of value for other universal healthcare funded systems, but also for different cancer sites with similar challenges.

Finally, patients treated within multidisciplinary teams receive better care and have better outcomes than those treated without MDT expertise. MDT discussions are mandatory in the UK NHS, yet some variability has been described particularly at recurrence [66]. Our work further reinforces the need for patient care to be delivered within MDTs [67,68,69].

Our work complements worldwide efforts to achieve universal health coverage (UHC) with all people receiving high standard care, without experiencing financial hardship and without having to bear the often-detrimental effects of the postcode lottery [27]. Making progress towards UHC has been highlighted by the UN Sustainable Development Goals and WHO’s Thirteenth General Programme of Work (GPW13), as a priority policy for countries and global institutions [27]. A crucial step to achieve this is to set and validate objective, easily-assessed metrics, i.e., QPIs, to measure whether health services are aligned with countries’ health profiles and optimal standards, and whether they are of sufficient quality to produce health gains for populations of all ages and ethnicities [27].

## 6. Limitations

Validation of these QPI in the long-term NHS funded national audit will be important to improve outcomes in ovarian cancer. In order for other countries to know whether they would be able to follow these QPI and interventions, harmonisation of data collected will be critical. In the UK, national datasets routinely collect person data, all hospital admission data and key cancer parameters (e.g., Stage, histology). Data collection is notoriously difficult across many precincts in cancer care and health in general. Sufficient data would be required in order to allow averages to be adjusted for age/stage/histology type/comorbidity/deprivation in order for hospitals/jurisdictions to not be disadvantaged. This is extremely important in order for different jurisdictions to be willing to participate and to be benchmarked.

A limitation of the OCAFP is that we are unable to shed light on access to diagnosis in primary care. This is likely to be very relevant, particularly in women not receiving any anticancer treatment, due to delayed diagnosis and poor performance status at presentation. However, the National Ovarian cancer feasibility pilot report on short term mortality from Ovarian cancer does not find a correlation between primary care/family doctor to patient ratio. Future national efforts will focus on greater understanding of the primary care diagnostic pathway and how this can be addressed [70].

We acknowledge that no QPI have been recommended for Homologous recombination deficiency testing as data collection efforts are still ongoing [71].

Future iterations of the BGCS QPI will need to generate this based on nationally available data.

## 7. Conclusions

We propose here, well-defined metrics and QPIs for the improvement of ovarian cancer management. Even though developed within the UK, learning acquired from this process will be of benefit for other universal healthcare funded systems. The principles are consistent and clear: data-driven evidence; gathered from comprehensive, routinely-collected, prospective registries; use of national populations as denominators; and considering the entire diagnostic and therapeutic pathway. Key to the success of such an approach is leadership by clinicians, engagement of the broader clinical community, and collaboration with patient advocates, under the umbrella of national societies. Reporting performance against metrics in public domain whilst allowing for near-time (within 2 years) feedback to individual service providers, will, we hope, create a virtuous feedback loop that promotes improvement in outcomes that are meaningful to patients and communities. Health systems need to explore mechanisms to deliver the service-level changes that may arise from implementing these QPIs.

## Figures and Tables

**Figure 1 cancers-15-00337-f001:**
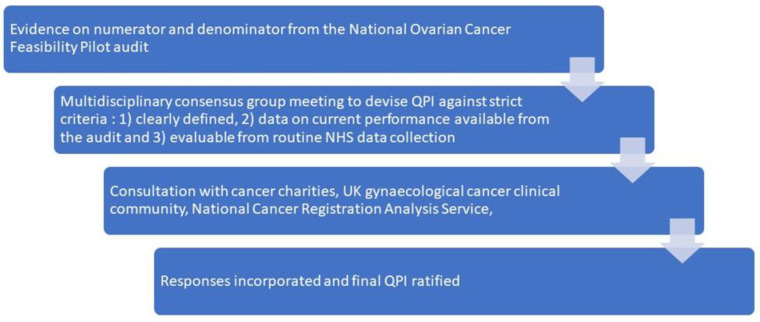
Production of BGCS Quality performance indicators.

**Table 1 cancers-15-00337-t001:** Summary of QPI including numerators and denominators for QPI measurement.

QPI	Number	Indicator	Target	Description and Reporting
Patients to be discussed at diagnosis at a specialist MDT prior to a decision for treatment.	QPI 1	Process	Target 95%	Number of patients with ovarian cancer discussed at the MDT prior to a decision for definitive treatment. Numerator: Number of patients with ovarian cancer discussed at the MDT prior to a decision for definitive treatment. Denominator: All patients diagnosed with ovarian cancer. Exclusions: Borderline ovarian tumours. Reportable by: Hospital Trust; Integrated Cancer System; and Cancer Alliance. Reported on: CancerStats2 and in public domain reports
Patients diagnosed with Stage II-IV or unstaged Ovarian cancer to receive anticancer treatment of any type.	QPI 2	Process	Target 80%	Numerator: patients with stage II-IV or unstaged ovarian cancer receiving anticancer treatment. Denominator: All patients with stage II-IV or unstaged ovarian cancer diagnosed. Exclusions: Borderline ovarian tumours. Reportable by: Hospital Trust; Integrated Cancer System; and Cancer Alliance. Reported on: CancerStats2 and in public domain reports. Data in public domain to be reported adjusted for age and deprivation.
Patients with Stage II-IV/unstaged ovarian cancer to receive cytoreductive surgery.	QPI 3	Structural	Minimum target 55%; Optimal target 70%.	Numerator: patients receiving primary surgery or delayed debulking surgery after neoadjuvant chemotherapy. Denominator: All Stage II-IV/unstaged patients with ovarian cancer. Exclusion: Borderline ovarian tumours. Reportable by: Hospital Trust; Integrated Cancer System; and Cancer Alliance. Reported on: CancerStats2 and in public domain reports. Data in public domain to be reported adjusted for age and deprivation.
Patients with ovarian cancer should have recording of FIGO stage, WHO performance status, at diagnosis.	QPI 4a	Outcome	Target 95%	For performance status and stage. Numerator: All patients with ovarian cancer discussed at MDT. Denominator: All patients with ovarian cancer. Exclusion: borderline ovarian tumours. Reportable by: Hospital Trust; Integrated Cancer System; and Cancer Alliance. Reported on: CancerStats2 and in public domain reports.
Patients with ovarian cancer undergoing primary or interval debulking surgery should have recording of residual disease.	QPI 4b	Outcome	Target 95%	Numerator: All patients with ovarian cancer undergoing surgery. Denominator: All patients with ovarian cancer. Exclusion: borderline ovarian tumours. Reportable by: Hospital Trust; Integrated Cancer System; and Cancer Alliance. Reported on: CancerStats2 and in public domain reports.
Patients with non-mucinous epithelial ovarian cancer on histology to be tested for germline BRCA1/2 testing.	QPI 5a	Outcome	Target 90%	Numerator: All patients with non-mucinous epithelial ovarian cancer histology, including those with missing histology or unspecified histology undergoing testing for germline BRCA1/2. Denominator: All patients with non-mucinous epithelial ovarian cancer histology, including those with missing histology or unspecified histology. Exclusion: borderline ovarian tumours, mucinous epithelial ovarian cancers. Reportable by: Hospital Trust; Integrated Cancer System; and Cancer Alliance. Reported on: CancerStats2 and in public domain reports.
Patients with advanced high grade serous and clear cell cancer on histology to be tested for tumour BRCA1/2 testing.	QPI 5b	Outcome	Target 90%	Numerator: All patients with Stage III-IV/unstaged high grade serous or clear cell epithelial ovarian cancer histology, including those with missing histology or unspecified histology tested for tumour BRCA1/2. Denominator: All patients with Stage III-IV/unstaged high grade serous or clear cell epithelial ovarian cancer histology, including those with missing histology or unspecified histology. Exclusion: borderline ovarian tumours. Stage 1–2 cancer, histology types other than high grade serous or clear cell epithelial ovarian cancer. Reportable by: Hospital Trust; Integrated Cancer System; and Cancer Alliance. Reported on: CancerStats2 and in public domain reports.
Patients to be enrolled into an NCRI portfolio study at diagnosis.	QPI 6	Structural	Minimum Target 5%	Numerator: number of patients with ovarian cancer discussed at the MDT recruited into a NCRI portfolio study. Denominator: All patients diagnosed with ovarian cancer. Exclusions: borderline ovarian tumours. Reportable by: Hospital Trust; Integrated Cancer System; and Cancer Alliance. Reported on: Public domain reports.

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
