# Peer review of "British Gynaecological Cancer Society Recommendations for Evidence Based, Population Data Derived Quality Performance Indicators for Ovarian Cancer"

_cancers, 2023, doi:10.3390/cancers15020337_

Round 1

Reviewer 1 Report

Sundar and colleagues presented a manuscript aimed at proposing some recommendations to improve the management and overall survival of ovarian cancer patients. For this purpose, the authors analyzed the data derived from several quality performance indicators outlining possible strategies to improve such indicators. Overall, the manuscript may be helpful for clinicians, however, there are some issues that the authors have to address before publication:

1) The Description of some chapters should be more fluent. It is suggested to summarize the content of chapters 1 and 2;

2) The authors have to provide a schematic representation of the QPIs evaluated and the methodologies used for the evaluation of QPIs. This would be helpful to readers for a faster understanding of the contents of the manuscript;

3) Please clarify the following sentence: “Analysis of early mortality at the OCAFP demonstrates that patients diagnosed at cancer units have about 10% lower survival than patients diagnosed at cancer centres.1“. Did the authors mean that patients treated in general hospitals have lower survival compared to patients diagnosed at cancer centers? Please, clarify;

4) Please use the same page layout for all QPIs (i.e. 3.3 QPI 3, 3.3.1 Rationale, 3.3.2 Best practice solutions);

5) I strongly believe that it is necessary to refer to international gynecological guidelines to improve the quality of health assistance in UK. In the Discussion section, please briefly mention the guidelines proposed by ESMO, ASCO and the main argument of the Consensus Conferences on this topic;

6) In the Discussion section, the authors have to further emphasize the importance of a multidisciplinary approach to the management of ovarian cancer patients. It was recently demonstrated how cancer centers where multidisciplinary teams operate have better performance in terms of curative surgery or prolonged patients' survival. Please add an in-depth description of this topic. For this purpose, please see:

- PMID: 34132354

- 10.1136/ijgc-2021-ESGO.339

- PMID: 35267603

- PMID: 34787913

Author Response

Reviewer 1

Sundar and colleagues presented a manuscript aimed at proposing some recommendations to improve the management and overall survival of ovarian cancer patients. For this purpose, the authors analyzed the data derived from several quality performance indicators outlining possible strategies to improve such indicators.

We thank the reviewer for their comments.

Overall, the manuscript may be helpful for clinicians, however, there are some issues that the authors have to address before publication:

  • The Description of some chapters should be more fluent. It is suggested to summarize the content of chapters 1 and 2;

Response. We sincerely apologise – we do not understand what is meant here. The manuscript has no chapters.

  • The authors have to provide a schematic representation of the QPIs evaluated and the methodologies used for the evaluation of QPIs. This would be helpful to readers for a faster understanding of the contents of the manuscript;

Response – We have now included a schematic of the development of QPI as Figure 1, line 292.The list of QPI and the methodology used is provided in Table 1.

3) Please clarify the following sentence: “Analysis of early mortality at the OCAFP demonstrates that patients diagnosed at cancer units have about 10% lower survival than patients diagnosed at cancer centres.1“. Did the authors mean that patients treated in general hospitals have lower survival compared to patients diagnosed at cancer centers? Please, clarify;

Response – We have added the sentence, line 323 ‘The reason for this is unclear and needs further research. One hypothesis is that poorer outcomes result from a longer diagnostic process, particularly in older, more frail patients. Future iterations of the National audit will investigate this further.

4) Please use the same page layout for all QPIs (i.e. 3.3 QPI 3, 3.3.1 Rationale, 3.3.2 Best practice solutions);

Response – we have done this

5) I strongly believe that it is necessary to refer to international gynecological guidelines to improve the quality of health assistance in UK. In the Discussion section, please briefly mention the guidelines proposed by ESMO, ASCO and the main argument of the Consensus Conferences on this topic;

Response – the importance of ESGO indicators is present in the manuscript lines, 238-241. We have further reiterated this in the Discussion, lines 662 - 668

6) In the Discussion section, the authors have to further emphasize the importance of a multidisciplinary approach to the management of ovarian cancer patients. It was recently demonstrated how cancer centers where multidisciplinary teams operate have better performance in terms of curative surgery or prolonged patients' survival. Please add an in-depth description of this topic. For this purpose, please see:

- PMID: 34132354

- 10.1136/ijgc-2021-ESGO.339

- PMID: 35267603

- PMID: 34787913

Response We have added a section on the vital importance of multidisciplinary approaches to the care of the patient in Discussion, lines 675-678.  MDTs are routine care in the UK.

Reviewer 2 Report

The authors, representing the British Gynaecological Cancer Society, present a quality management program to improve national-wide results in ovarian cancer treatment. Although this paper is exclusively based on expert opinion, the decisions are sufficiently supported by previous (correctly referenced results). The paper is well written and will be of great interest to any other efforts to improve the care of ovarian cancer patients.

The epidemiology overview in Section 1 could be completed by citing a newer CONCORD-3 study published in The Lancet, 2018, 391(10125)pp. 1023-1075, DOI: 10.1016/S0140-6736(17)33326-3.

Author Response

Reviewer 2

The authors, representing the British Gynaecological Cancer Society, present a quality management program to improve national-wide results in ovarian cancer treatment. Although this paper is exclusively based on expert opinion, the decisions are sufficiently supported by previous (correctly referenced results). The paper is well written and will be of great interest to any other efforts to improve the care of ovarian cancer patients.

The epidemiology overview in Section 1 could be completed by citing a newer CONCORD-3 study published in The Lancet, 2018, 391(10125), pp. 1023-1075, DOI: 10.1016/S0140-6736(17)33326-3.

Response – we thank the reviewer for their supportive comments. We have added the CONCORD reference to the Background, line numbers 173-174

Reviewer 3 Report

SUMMARY:

This paper sets out recommendations and quality performance indicators (QPI) for ovarian cancer by the British Gynaecological Cancer Society. This is an important step forward given the poor survival from this disease among women in the UK compared with other countries. 

GENERAL COMMENTS:

QPIs are evidence-based and clearly defined.

The authors importantly specify that more research is needed to understand why some patients are not receiving treatment, and which the additional information outlined will facilitate.

QPI 5a/b: For greater context, how is germline BRCA1/2 testing paid for in England? If it is at the patient’s expense this is an obvious barrier to meeting the target. If absorbed by the healthcare system, will this (and other) approaches outlined be sustainable?

MINOR COMMENTS:

Document formatting is inconsistent, e.g., QPI 1 provides 3.1.1 Rationale and 3.1.2 Best practice solutions, while QPI 2 provides 3.2.1 Rationale and no numbering for Best practice.

Additional spaces between words are included throughout the paper.

Line 305: remove “to contrast”

Line 379: “for” instead of “doe”

Author Response

Reviewer 3

We thank the reviewer for their supportive comments

GENERAL COMMENTS:

The authors importantly specify that more research is needed to understand why some patients are not receiving treatment, and which the additional information outlined will facilitate.

QPI 5a/b: For greater context, how is germline BRCA1/2 testing paid for in England? If it is at the patient’s expense this is an obvious barrier to meeting the target. If absorbed by the healthcare system, will this (and other) approaches outlined be sustainable?

Response – Germline BRCA1/2 is provided by the National Health service and is free to the patient. We have added this to the manuscript in page 11, line 484. Sustainability of quality health care in the context of rising costs is critical; in the UK new intervention or diagnostics are usually assessed for cost effectiveness by an arm’s length body called National Institute of Health and Care Excellence ( NICE) and only implemented into routine health care if found to be cost effective.

MINOR COMMENTS:

Document formatting is inconsistent, e.g., QPI 1 provides 3.1.1 Rationale and 3.1.2 Best practice solutions, while QPI 2 provides 3.2.1 Rationale and no numbering for Best practice. Additional spaces between words are included throughout the paper.

Line 305: remove “to contrast”

Line 379: “for” instead of “doe”

Response – we have addressed formatting in the document and removed these errors.

Round 2

Reviewer 1 Report

Dear Authors thank you for your reply. Almost all of my previous comments have been properly addressed. However, there are still some minor revisions that you have to address before publication.

In my previous comment No. 1, I meant The chapter 1 "Background" and Chapter 2 "Learning from other health systems: the case for evidence-based, population data 144 derived, routinely evaluated QPIs";

You correctly answered to my previous comment No. 6, however, you should provide supporting references about the importance of multidisciplinary approach for patients with gynecological cancer (see some of the references previously suggested)

Author Response

In my previous comment No. 1, I meant The chapter 1 "Background" and Chapter 2 "Learning from other health systems: the case for evidence-based, population data 144 derived, routinely evaluated QPIs";

Response - Thank you very much for your time reviewing the manuscript again. We agree that summarising Chapters 1 and 2 helps to present the argument clearly. We have attempted this - please let us know if this is satisfactory

You correctly answered to my previous comment No. 6, however, you should provide supporting references about the importance of multidisciplinary approach for patients with gynecological cancer (see some of the references previously suggested)

Response - references are now inserted. 

Many thanks again and Merry Christmas!
